# Delineating MYC-Mediated Escape Mechanisms from Conventional and T Cell-Redirecting Therapeutic Antibodies

**DOI:** 10.3390/ijms252212094

**Published:** 2024-11-11

**Authors:** Anna Vera de Jonge, Tamás Csikós, Merve Eken, Elianne P. Bulthuis, Pino J. Poddighe, Margaretha G. M. Roemer, Martine E. D. Chamuleau, Tuna Mutis

**Affiliations:** 1Department of Hematology, Amsterdam UMC Location Vrije Universiteit, 1081 HV Amsterdam, The Netherlands; a.dejonge1@amsterdamumc.nl (A.V.d.J.);; 2Cancer Biology and Immunology, Cancer Center Amsterdam, 1081 HV Amsterdam, The Netherlands; 3Department of Clinical Human Genetics, Amsterdam UMC Location Vrije Universiteit, 1081 HV Amsterdam, The Netherlands

**Keywords:** MYC, MYC inhibition, B-cell malignancies, lymphoma, CRISPR-Cas9, antibody therapy, T cell-redirecting antibodies, immunotherapy

## Abstract

In B-cell malignancies, the overexpression of MYC is associated with poor prognosis, but its mechanism underlying resistance to immunochemotherapy remains less clear. In further investigations of this issue, we show here that the pharmacological inhibition of MYC in various lymphoma and multiple myeloma cell lines, as well as patient-derived primary tumor cells, enhances their susceptibility to NK cell-mediated cytotoxicity induced by conventional antibodies targeting CD20 (rituximab) and CD38 (daratumumab), as well as T cell-mediated cytotoxicity induced by the CD19-targeting bispecific T-cell engager blinatumomab. This was associated with upregulation of the target antigen only for rituximab, suggesting additional escape mechanisms. To investigate these mechanisms, we targeted the *MYC* gene in OCI-LY18 cells using CRISPR-Cas9 gene-editing technology. CRISPR-Cas9-mediated *MYC* targeting not only upregulated CD20 but also triggered broader apoptotic pathways, upregulating pro-apoptotic PUMA and downregulating anti-apoptotic proteins BCL-2, XIAP, survivin and MCL-1, thereby rendering tumor cells more prone to apoptosis, a key tumor-lysis mechanism employed by T-cells and NK-cells. Moreover, MYC downregulation boosted T-cell activation and cytokine release in response to blinatumomab, revealing a MYC-mediated T-cell suppression mechanism. In conclusion, MYC overexpressing tumor cells mitigated the efficacy of therapeutic antibodies through several non-overlapping mechanisms. Given the challenges associated with direct MYC inhibition due to toxicity, successful modulation of MYC-mediated immune evasion mechanisms may improve the outcome of immunotherapeutic approaches in B-cell malignancies.

## 1. Introduction 

As one of the most important transcription factors in cells, c-MYC (hereafter MYC) is involved in cell development and regulates cellular functions, including cell cycle control, cell metabolism, proliferation, differentiation and apoptosis [1,2]. Once deregulated, MYC acts as a proto-oncogene contributing to tumorigenesis by increasing cell proliferation and inhibiting cell differentiation [3,4]. *MYC*, located on chromosome 8q24.21, is overexpressed in up to 45% of solid tumors [5] and translocated in up to 15% of diffuse large B-cell lymphomas (DLBCL) [6] and multiple myeloma (MM) [7] patients, and in 95–100% of all Burkitt lymphoma (BL) patients [8,9]. Tumors overexpressing MYC often display resistance to chemotherapy, leading to higher incidences of relapse and elevated mortality rates [10,11,12,13,14]. Successful immunotherapeutic approaches for lymphoma, such as the anti-CD20 antibody rituximab in combination with chemotherapy, are also less effective in high-grade B-cell lymphomas displaying *MYC* rearrangements [14,15,16], suggesting the involvement of MYC in tumor escape from novel immunotherapeutic approaches. Supporting this idea, combining MYC inhibitors with anti-PD1 therapy effectively reduced tumor volumes in recent murine studies of prostate and pancreatic cancer [17,18]. While such evidence points to a significant contribution of MYC to immunotherapy resistance [19], the exact impact of MYC on the efficacy of immunotherapeutic approaches and possible mechanisms thereof remains largely unexplored. 

To gain more insight into this, we used synthetic MYC inhibitor 10058-F4 in a relatively broad panel of MYC overexpressing cell lines and primary tumor cells from B-cell malignancies, including DLBCL, BL and MM. We tested the susceptibility of all MYC downregulated cells against effective and clinically approved conventional antibodies targeting CD20 (rituximab) and CD38 (daratumumab) and a bispecific T-cell engager targeting CD19 (blinatumomab). We generated mutated MYC alleles in OCI-LY18 cells using CRISPR-Cas9 gene-editing technology and subsequently repaired mutated *MYC* coding sequences by CRISPR-Cas9-mediated homologous recombination to confirm the direct involvement of MYC in escape from conventional and T cell-redirecting antibodies. Our results reveal the involvement of MYC in tumor escape from NK cell- and T cell-mediated cytotoxicity through diverse, non-overlapping mechanisms. The results provide novel insights to improve the efficacy of immunotherapeutic approaches for MYC-driven malignancies.

## 2. Results

### 2.1. MYC Downregulation by Synthetic MYC Inhibitor 10058-F4 Sensitizes Lymphoma Cells Towards Rituximab-Mediated Antibody-Dependent Cellular Cytotoxicity Through CD20 Upregulation 

To study the impact of *MYC* on conventional and T cell-redirecting therapeutic antibodies, we first inhibited MYC in an extensive panel of DLBCL, BL and MM cell lines using the small molecule 10058-F4. This MYC inhibitor specifically inactivates MYC, but not its down- or upstream products, by inhibiting the dimerization of MYC with MYC-associated factor X (MAX) [20]. 

First, we confirmed decreased MYC protein expression at concentrations of 25–150 μM in DLBCL cell lines OCI-LY18, WSU-DLCL2, OCI-LY7 and SU-DHL-6; in BL cell lines Daudi and Raji; and in MM cell lines MM1.S and RPMI8226 (Figure 1A and Appendix A). 10058-F4 inhibited cell proliferation at concentrations of 100–150 μM (Appendix A). Rituximab as a single agent induced moderate lysis in the majority of the cell lines (Appendix A). In cytotoxicity assays, we pre-incubated tumor cells with 10058-F4 and washed it away to avoid toxicity on immune effector cells. The pre-incubation of DLBCL cell lines OCI-LY18, WSU-DLCL2, OCI-LY7 and BL cell lines Daudi and 10058-F4 synergistically improved rituximab-mediated tumor cytotoxicity (Figure 1B,C). Additive effects were observed in SU-DHL6 and Raji. While the MYC protein expression level was not associated with CD20 expression in a selection of DLBCL cell lines (Appendix A), the increased rituximab-mediated antibody-dependent cellular cytotoxicity (ADCC) was associated with increased CD20 surface expression (Figure 1D), pointing to a MYC-dependent regulation of CD20 expression, as previously suggested [21].

### 2.2. MYC Downregulation by Synthetic MYC Inhibitor 10058-F4 Sensitizes Lymphoma Cells Towards Daratumumab-Mediated and T Cell-Redirected Cytotoxicity Independent of Target Antigen Expression

After showing enhanced rituximab-mediated ADCC, we tested whether the efficacy of the monoclonal anti-CD38 antibody daratumumab and the CD3xCD19 bispecific T-cell engager blinatumomab could be improved by MYC downregulation. MYC downregulation by 10058-F4 also synergistically enhanced daratumumab-mediated ADCC in both MM cell lines (Figure 1E). In DLBCL cell lines, 10058-F4 did not affect CD38 expression (Appendix A).

The pre-incubation with 10058-F4 synergistically enhanced blinatumomab-mediated T cell-dependent cytotoxicity in WSU-DLCL2 and showed additivity in other DLBCL cell lines (Figure 1F). Remarkably, the enhanced blinatumomab-mediated cytotoxicity in MYC downregulated clones was not associated with the upregulation of target antigen expression (Figure 1G and Appendix A). Hence, these data indicated that MYC-associated escape mechanisms from immunotherapy extend beyond simple antigen modulation on the cell surface.

### 2.3. MYC Targeting by CRISPR-Cas9 Gene Editing 

To study the effects of MYC targeting and to investigate the mechanisms underlying the increased susceptibility to conventional and T cell-redirecting therapeutic antibodies, we next targeted the *MYC* gene in OCI-LY18 using CRISPR-Cas9 gene-editing technology. OCI-LY18 overexpresses MYC due to a pathogenic translocation of the *MYC* gene with the immunoglobulin heavy chain enhancer gene (*IGH*, t(8;14)(q24;q32)). We observed that all viable *MYC-targeted* daughter clones retained at least a *MYC* open reading frame, confirming that a complete loss of function of *MYC* is detrimental to cell proliferation (Table 1). We selected 10 clones which retained a single MYC ORF for further evaluation to overcome any possible bias introduced by clonal heterogeneity. Immunoblotting showed MYC protein downregulation in all clones (Figure 2A). We, therefore, refer to these clones as “MYC downregulated”. As expected, the proliferation rate of these MYC downregulated clones was lower compared to the parental OCI-LY18 wild type (WT) cells after 2–3 days (Appendix A). Given the association between MYC and decreased expression of MHC molecules, we explored the phenotype of our clones. In agreement with other reports [22,23], MYC downregulated clones showed upregulation of HLA-DR (Appendix A). Data on the association between the expression of intracellular MYC and cell surface immune checkpoint molecules are conflicting in lymphoma [24]. In accordance with a previous study [25], we observed no changes in PD-L1 in MYC downregulated clones compared to parental OCI-LY18 WT cells. We observed no changes in CD47 expression (Appendix A).

### 2.4. Increased Rituximab-Mediated ADCC and CD20 Surface Expression Are a Direct Consequence of MYC Downregulation

After the initial characterization of MYC downregulated cells, we tested their susceptibility against rituximab-mediated antibody-dependent cellular cytotoxicity (ADCC). While the introduction of Cas9 did not have any effect on rituximab-mediated ADCC or target antigen expression (Appendix A), rituximab-mediated ADCC was increased in nine out of ten MYC downregulated clones (Figure 2B). Again, the increased rituximab-mediated ADCC was associated with increased CD20 surface expression (Figure 2C,D), pointing to a MYC-dependent regulation of CD20 expression. In light of current antibody development for the treatment of DLBCL, we evaluated the expression of other pan-B-cell targetable antigens CD37, CD79b and CD22. Interestingly, CD37, which co-localizes with CD20 [26], was also upregulated in MYC downregulated clones in a similar manner as CD20 (Appendix A). The expression of CD79b and CD22 was not affected (Appendix A).

To confirm the direct involvement of MYC in the increased rituximab-mediated ADCC and CD20 expression, we repaired the mutated *MYC* coding sequence in the representative MYC downregulated clone #1 by homologous recombination (Table 2). Several successfully “MYC-repaired” subclones displayed a similar proliferation rate, a similar sensitivity towards rituximab-mediated cytotoxicity and similar CD20 expression levels as OCI-LY18 WT (Appendix A). We selected two representative repaired subclones (“repair 1” and “repair 2”) for further experiments. These subclones showed similar MYC protein expression compared to parental OCI-LY18 WT cells (Figure 2E). In both repaired subclones, rituximab-mediated cytotoxicity and CD20 expression were reversed towards the parental OCI-LY18 WT phenotype (Figure 2F,G), confirming that the increase in rituximab-mediated ADCC and CD20 surface expression were a direct consequence of MYC downregulation. 

### 2.5. Increased Sensitivity Towards Conventional and T Cell-Redirected Cytotoxicity Beyond Target Antigen Expression Following MYC Downregulation 

We also tested whether the efficacy of the monoclonal anti-CD38 antibody daratumumab and the CD3xCD19 bispecific T-cell engager blinatumomab could be improved by CRISPR-Cas9-mediated MYC downregulation. Similar to the rituximab setting, we first confirmed that Cas9 transduction did not influence daratumumab-mediated cytotoxicity (Appendix A). Interestingly, daratumumab-mediated ADCC was enhanced in the same MYC downregulated clones as parental OCI-LY18 WT cells (Figure 3A) as well as blinatumomab-mediated T cell-dependent cytotoxicity in a selection of representative MYC downregulated clones tested (Figure 3B). Daratumumab- and blinatumomab-mediated cytotoxicity was completely reversed in MYC-repaired subclones, confirming the direct involvement of MYC (Figure 3C,D). Cas9 transduction did not influence CD38 or CD19 expression in parental OCI-Y18 WT cells (Appendix A), and MYC downregulated clones displayed equal or reduced expression levels of target antigens for daratumumab (CD38) and blinatumomab (CD19) compared to parental OCI-LY18 WT cells (Figure 3E). In addition, successfully MYC-repaired subclones displayed a similar CD19 and CD38 expression as parental OCI-LY18 WT cells (Appendix A and Figure 3F,G).

Single-nucleotide polymorphism (SNP)-based genomic array profiling revealed that all MYC downregulated clones had more or less the same chromosomal gains and losses as were present in OCI-LY18 WT cells and OCI-LY18-Cas9 cells. Clones #3, #5 and #9 showed additional chromosomal abnormalities, but these seemed to be nonspecific and not related to the CD20 locus on chromosome 11q12.2 or the other target loci for CD38 (4p15.32) or CD19 (16p11.2) (Appendix A). Therefore, the limited degree of genomic heterogeneity within the clones did not seem to affect the target antigen expression of CD20, CD38 and CD19.

These results indicate that modulating target antigen expression was not the only mechanism for the evasion of immunotherapeutic approaches in MYC overexpressing tumor cells. We next aimed to further delineate these additional mechanisms.

### 2.6. Increased Cytotoxicity in MYC Downregulated Cells Is Associated with Downregulation of Anti-Apoptotic Proteins

In MM cells, the upregulation of anti-apoptotic pathways can mitigate NK cell- and T cell-mediated cytotoxicity [27,28,29]. To evaluate whether MYC could blunt the efficacy of immunotherapeutic approaches by modulation of apoptosis, we measured the expression of the most important apoptosis regulatory proteins BCL-2, MCL1, survivin and XIAP, and pro-apoptotic proteins PUMA and BIM in the representative clone #1 and its MYC-repaired subclones. 

The MYC downregulated clone #1 displayed lower expression levels of anti-apoptotic proteins XIAP, survivin, MCL-1 and BCL-2, and a higher expression level of the pro-apoptotic PUMA (Figure 4A) compared to parental OCI-LY18 WT cells. The direct involvement of MYC in these events was confirmed by the restored expression of XIAP, PUMA and, to a lesser extent, survivin and MCL-1 in both MYC-repaired subclones (Figure 4A). The strong correlation between enhanced cytotoxicity and the downregulation of anti-apoptotic pathways in MYC downregulated clones strongly suggests that the upregulation of anti-apoptotic pathways serves as an additional mechanism through which MYC overexpressing tumor cells evade immunotherapeutic approaches.

### 2.7. MYC Downregulation Improves the Activation State of T-Cells Engaged by Blinatumomab

We next evaluated whether lymphoma-intrinsic MYC could suppress the activity of effector T-cells by comparing their cytokine release after T-cell engager blinatumomab-mediated cytotoxicity against parental OCI-LY18 WT versus MYC downregulated cells.

The increased blinatumomab-mediated cytotoxicity of MYC downregulated clones was accompanied by significantly higher levels of IL-2, IFN-γ, IL-10 and TNF-α secretion (Figure 4B). The Granzyme B, IL-6 and IL-4 secretion was also higher in some of the clones (Figure 4C). Overall, the increased cytokine secretion confirmed that MYC overexpression had a suppressive impact on effector T-cells. Additionally, we investigated the tumor cell surface expression of molecules involved in T cell-mediated cytotoxicity. Although the expression of HVEM and galectin-9 was increased in MYC downregulated clones (Appendix A), CD155 (ligand of TIGIT) was decreased, and the expression of adhesion and costimulatory molecules CD58 and CD80 was increased (Appendix A). These findings point towards the suppression of T-cell activation as a third mechanism of MYC-mediated escape from T cell-based immunotherapeutic approaches. 

### 2.8. MYC Inhibition by Synthetic MYC Inhibitor 10058-F4 Boosts Antibody Therapy in B-Cell Malignancies Despite Tumor Heterogeneity

Finally, we evaluated primary tumor cell samples directly obtained from DLBCL and MM patients using the synthetic MYC inhibitor 10058-F4. These lymph node and bone marrow samples also contained the necessary immune effector cells. Therefore, to mitigate the toxic effects of 10058-F4 on T-cells and NK-cells, we directly added 10058-F4 to the assay for a shorter time period (16 h) at concentrations that were non-toxic for NK-cells (12.5–50 μM, Figure 5A). As a single agent, 10058-F4 induced moderate cytotoxicity in DLBCL and MM cells at these concentrations (Figure 5B). Rituximab- and daratumumab-mediated cytotoxicity was moderate to good (Figure 5C). The combination of 10058-F4 (25 μM) with rituximab or daratumumab (100 ng/mL) showed synergism in 7/15 samples (Figure 5D), indicating the possibility to improve the efficacy of immunotherapeutic approaches in primary B-cell malignancies through MYC inhibition. 

Overall, these results indicate that successful MYC inhibition can improve the efficacy of immunotherapeutic antibodies in several B-cell malignancies, albeit with some heterogeneity.

## 3. Discussion

We demonstrated that MYC mitigates the efficacy of clinically available conventional and T cell-redirecting antibodies targeting different surface antigens (CD20, CD38 and CD19) using pharmacological and genetic approaches of MYC targeting in vitro and ex vivo in various B-cell malignancies. We revealed that MYC can blunt the efficacy of immunotherapeutic approaches via target antigen downregulation in the case of CD20, but also via induction of tumor intrinsic resistance towards NK cell- and T cell-mediated cytotoxicity by upregulating anti-apoptotic pathways and suppressing T-cell activation.

To investigate a broad range of cell lines and primary cells derived from B-cell malignancies, we used the synthetic MYC inhibitor 10058-F4. To date, the use of various other agents has been studied to target MYC-driven tumors [30,31]. Many of these agents also modulate other pathways and cellular processes, inducing off-target effects, making it difficult to specifically study the effect on MYC. We selected 10058-F4 over alternatives, such as BET bromodomain inhibitors or kinase inhibitors, because 10058-F4 specifically inactivates MYC by binding directly to MYC’s basic-helix-loop-helix-leucine-zipper domain [20], specifically at the Helix 2-ZIP junction (residues 402–412) [32]. Subsequently, MYC’s directed transcription is disabled, and MYC mRNA and protein levels are downregulated, either due to a loss of acetylation or due to a let-7-mediated negative feedback loop [33]. Hence, the effects observed here after treatment with 10058-F4 can be directly attributed to MYC. 

To avoid the negative side effects of MYC inhibition on peripheral blood immune effector cells and to explore the mechanisms behind the increased susceptibility to therapeutic antibodies, we additionally targeted MYC using CRISPR-Cas9 gene-editing technology. CRISPR-Cas9 efficiently causes stable short frameshift insertion–deletions into protein coding target genes that allow for a reduced risk of off-target effects [34] whilst specifically examining the effect of *MYC* modulation. The standard approach herein is to obtain single cell-derived clones, which may originally display subtle genetic or phenotypic differences. To outbalance any bias introduced by genetic heterogeneity, we analyzed 10 *MYC-targeted* clones. We observed limited and nonspecific genetic heterogeneities. Nine out of ten clones exhibited similar results that were reversed after the successful repair of *MYC* in a number of subclones. These results convincingly demonstrated that specific *MYC* targeting, but not off-target effects or inter-clonal genetic heterogeneity, was responsible for the observed results. 

Among the mechanisms that we observed by which MYC mitigates the efficacy of rituximab, daratumumab and blinatumomab, only the MYC-mediated downregulation of CD20 was described earlier [21,35]. It was previously suggested that MYC directly represses the promotor of the *MS4A1* gene, encoding CD20 [21,35], and via induction of the MS4A1/CD20-targeting miR-222 [21].

The upregulation of CD20, but not CD38 or CD19, expression in MYC downregulated clones points to the relative importance of MYC-mediated intracellular escape mechanisms against NK-cells and T-cells beyond target antigen modulation. Interestingly, the efficacy of all immunotherapeutic approaches was similarly increased in the nine clones, indicating converging escape mechanisms against NK-cells and T-cells. To our knowledge, the MYC-mediated apoptosis deregulation and direct suppression of T-cell activation as escape mechanisms from immunotherapeutic approaches have not been demonstrated before.

Apoptosis induction is a key mechanism of NK cell- and T cell-mediated tumor cytotoxicity. Our results indicate that MYC can effectively modulate apoptotic proteins to induce immune escape from NK cell- and T cell-mediated cytotoxic machinery. MYC may be additionally involved in other processes relevant to immune-mediated tumor cytotoxicity, including inhibiting ferroptosis and necroptosis [36,37], contributing to evasion from anti-tumor immune responses via non-apoptotic mechanisms of programmed cell death.

The lymphoma-intrinsic MYC-mediated suppression of T-cell activation may occur via the increased secretion of immunosuppressive cytokines, inhibition of inflammatory signaling [38] or modulation of accessory molecules involved in immune synapses and the immunosuppressive tumor microenvironment. Investigating these immunosuppressive mechanisms associated with MYC is relevant for future studies. A major challenge in this context is the absence of proper in vivo models, as continuous overexpression or complete inhibition of MYC can be lethal for the host [39]. A tetracycline-regulated model of MYC-induced T-cell acute lymphoblastic leukemia has been developed [40]. However, in vivo models are often immunocompromised, which complicates the study of immunotherapeutic approaches. Novel in vivo models incorporating the human immune system and engrafted with patient-derived tumor xenografts [41] probably face challenges with high toxicity induced by direct MYC inhibitors.

MYC-mediated escape from currently used immunotherapeutic approaches has important clinical implications. First, it may explain why MYC overexpressing lymphomas are often refractory to R-CHOP immunochemotherapy. Second, it suggests that the treatment of MYC-driven malignancies could be improved by additional modulation of the MYC-mediated immune escape mechanisms. The challenge for the clinic is determining the optimal way to inhibit MYC. Despite widely available preclinical approaches to inhibit MYC [19,24], the early clinical use of direct MYC inhibitors did not proceed into clinical practice due to complicated drug development and undesired side effects [42]. MYC also exerts numerous physiological functions in NK-cells and T-cells [43,44,45]. Moreover, in our experimental approach, we have observed toxic effects of 10058-F4 on immune effector cells at concentrations exceeding 50 μM.

To translate our findings into clinical applicability, we propose to interfere with the mechanisms by which MYC overexpressing tumors escape from immunotherapeutic approaches. For instance, since MYC overexpressing lymphomas escape rituximab-mediated cytotoxicity by downregulating CD20, patients may benefit from agents that upregulate CD20, such as the histone deacetylase (HDAC) inhibitor valproate [46], the chemotherapeutic gemcitabine [47] or PIM kinase inhibitors [21]. Valproate and gemcitabine are clinically available and show beneficial results in DLBCL patients [48,49].

Alternatively, agents that downregulate anti-apoptotic proteins BCL-2, MCL-1, XIAP and/or survivin may be considered. For instance, venetoclax is an oral BCL-2 inhibitor that was considered a promising addition to R-CHOP, especially in lymphoma patients with MYC and BCL-2 overexpression [50]. However, the addition of venetoclax to the more intensive immunochemotherapeutic regimen EPOCH-R resulted in considerable toxicities [51]. Nevertheless, the fact that in our study the expression of anti-apoptotic BCL-2 was not restored in repaired subclones suggests that enhanced ADCC after MYC modulation is not solely influenced by BCL-2, highlighting the multifaceted mechanisms of MYC to escape from anti-tumor immune responses. Therefore, other agents that inhibit several anti-apoptotic proteins simultaneously may be more beneficial. The experimental drug FL118 downregulates MYC and the anti-apoptotic proteins survivin, MCL1 and XIAP [52,53], can modulate immune escape induced by the bone marrow microenvironment [28] and has shown promising pre-clinical activity in MM [27]. Of course, for each of these agents, careful studies are needed to determine an optimal dosage that has little or no toxic effects on NK-cells and T-cells. In addition, agents that improve the activity of NK-cells and T-cells could be advantageous. This idea is strengthened by recent findings that lenalidomide, an oral immunomodulatory drug that has stimulatory effects on NK-cells and T-cells and downregulates the MYC protein via cereblon modulation [54,55], is beneficial in addition to R-CHOP in patients with *MYC*-rearranged DLBCL [56].

In conclusion, our data demonstrate that MYC overexpression has a crucial role in escape from conventional and T cell-redirecting therapeutic antibodies. We discovered that underlying mechanisms involve the downregulation of target antigen expression, the upregulation of anti-apoptotic proteins and the suppression of T-cell activation. Interfering with these mechanisms may increase the success of immunotherapeutic approaches in patients with MYC overexpressing malignancies.

## 4. Materials and Methods

### 4.1. Primary Patient Samples and Peripheral Blood Mononuclear Cells from Healthy Donors

A detailed description of the acquisition of primary lymphoma, bone marrow cells and healthy donor peripheral blood mononuclear cells (PBMCs) is provided in the Appendix A. All patient samples were collected according to the *code of conduct for medical research* developed by the Council of the Federation of Medical Scientific Societies (FEDERA). 

### 4.2. Cell Lines

The DLBCL, BL and MM cell lines used and cell culture conditions are described in the Appendix A.

### 4.3. Antibodies and MYC Inhibitor

Rituximab (Celltrion, Zaventem, Belgium), daratumumab (Johnson and Johnson, New Brunswick, NK, USA), blinatumomab (Amgen, Tokyo, Japan) and the MYC inhibitor 10058-F4 (Selleckchem, Houston, TX, USA) were purchased commercially.

### 4.4. CRISPR-Cas9-Mediated Targeting of the MYC Open Reading Frame

Detailed information on DNA processing reagents, amplification, sequencing, generation and cloning of the sgRNA expression cassettes and lentivirus production is provided in the Appendix A.

The optimal single-guide RNA (sgRNA) target sites for the *MYC* DNA open reading frame (ORF) MYC-1 5′ CTTCGGGGAGACAACGACGG 3′ and MYC-2 5′ CTATGACCTCGACTACGACT 3′ were identified by design tools available on the GPP Web Portal (https://portals.broadinstitute.org/gpp/public/ accessed on 6 May 2019) [57,58]. Annealed primers encoding the sgRNA target sequence and additional bases to facilitate cloning were ligated into the pLCKO lentiviral backbone and subsequently sequenced. 

### 4.5. Production of Cas9-Expressing OCI-LY18 Cells

The Cas9-expressing lentivirus was provided by the Department of Oncogenetics of Amsterdam UMC, Amsterdam, The Netherlands. OCI-LY18 cells were transduced with this lentivirus, originally generated from the Lenti-Cas9-2A-Blast DNA vector in the presence of 8 µg/mL polybrene. Cells were selected by 10 µg/mL blasticidin (InvivoGen, San Diego, CA, USA) for at least 10 days. 

### 4.6. Generation of MYC-Mutated OCI-LY18-Cas9 Cells

OCI-LY18-Cas9 cells were transduced with lentivirus expressing either sgRNA MYC1 or sgRNA MYC2. Transduced cells were kept under selective pressure with 4 µg/mL puromycin (InvivoGen, San Diego, CA, USA) for at least a week. Single-cell clones were obtained by limiting dilution at a density of 0.4 cells per well in a 96-well plate. To facilitate clone expansion, 8000 irradiated (10 Gy) wild type OCI-LY18 cells were added as feeder cells in IMDM medium supplemented with 10% FCS and 1% penicillin/streptomycin (Invivogen, San Diego, CA, USA, 15140122) at 37 °C, 5% CO_2_ under a continuously selective pressure of 5 μg/mL blasticidin (InvivoGen, San Diego, CA, USA) and 4 g/mL puromycin (InvivoGen). 

Detailed information on the PCR and sequence validation of CRISPR-Cas9 *MYC* targeting is provided in the Appendix A.

### 4.7. Repair of MYC1-Mutation in Cells

The modified Alt-R^®^ homology direct repair (HDR) donor DNA template was obtained from Integrated DNA Technologies GmbH (IDT, München, Germany).

The bare nucleotide sequence of the HDR donor DNA template is 5′ CCCCGCCCCTGTCCCCTAGCCGCCGCTCCGGGCTCTGCTCTCCCTCTTACGTCGCTGTCACACCCTTCTCCCTTCGTGGAGATAATGATGGCGGTGGCGGGAGCTTCTCCACGGCCGACCAGCTGGAGA 3′.

Lentivirus expressing the sgRNA target sequence 5′ CTGCTCGCCCTCCTACGTTG 3′, designed to cut DNA just upstream of the *MYC*1 target site, was co-transfected with the Alt-R^®^ HDR donor DNA template (IDT) into *MYC*1-mutated OCI-LY18 Cas9-sgRNA-expressing cells. Briefly, a mixture of 82 µL Nucleofactor solution and 18 µL supplement (Amaxa Cell Line Nucleofactor Kit V, VCA-1003) containing 4 million cells, 4 µg vector, 1.2 µM Alt-R^®^ HDR Enhancer (IDT) and 2 µM repair template was prepared. After electroporation, using program B25 in the Nucleofector I device (Amaxa Biosystems, Lonza, Basel, Switzerland), cells were recovered in a 4 mL medium with 1.2 µM HDR Enhancer. The cells were allowed to recover for two days before putting them under selective pressure of 400 µg/mL geneticin (G418 ThermoFisher Scientific, Waltham, MA, USA, 10131-035) for two weeks. The transfection efficiency never exceeded 0.1%.

### 4.8. Cytotoxicity Assays

Tumor (target) cells were pre-incubated with serial dilutions of indicated agents (rituximab (0–1000 ng/mL), daratumumab (0–10.00 ng/mL) or blinatumomab (0–10 ng/mL)) for 20–30 min on a rocking platform at room temperature. Then, thawed human PBMCs were added at the indicated effector (PBMC) to target (E:T) ratios to induce antibody-dependent cellular cytotoxicity (ADCC; rituximab, daratumumab) for 16 h, or T cell-mediated lysis (blinatumomab) for 24 h at 37 °C, 5% CO_2_. Thereafter, the surviving tumor cells were enumerated by standard quantitative flow cytometry (Appendix A) or bioluminescent imaging-based assay (a detailed description is provided in the Appendix A).

### 4.9. Supplemental Methods

A detailed description of other materials, including antibodies used for western blot (Appendix A), and methods is provided in the Appendix A.

### 4.10. Statistical Analysis

The data were analyzed and visualized using GraphPad Prism 8.2.1 and FCS Express Flow Cytometry Software (De Novo Software versions 06.0025 and 7.14.0020). The graphs represent mean values ± standard error of the mean (SEM). The statistical significance between the two measurements was determined using the paired Student’s *t*-test. 

Where indicated, observed lysis levels [O] obtained after a combination treatment were compared to the expected lysis levels [E], which were calculated with the assumption that the combinatorial effect is achieved by additive effects according to the BLISS model: % expected lysis = ((% lysis 10058-F4/100) + (% lysis antibody/100)) − ((% lysis 10058-F4/100) × (% lysis antibody/100)) × 100. Significantly higher observed lysis values (determined using the paired Student’s *t*-test) indicated synergism, and non-significant differences indicated additivity. * *p* < 0.05; ** *p* < 0.01; *** *p* < 0.001; **** *p* < 0.0001.

## Figures and Tables

**Figure 1 ijms-25-12094-f001:**
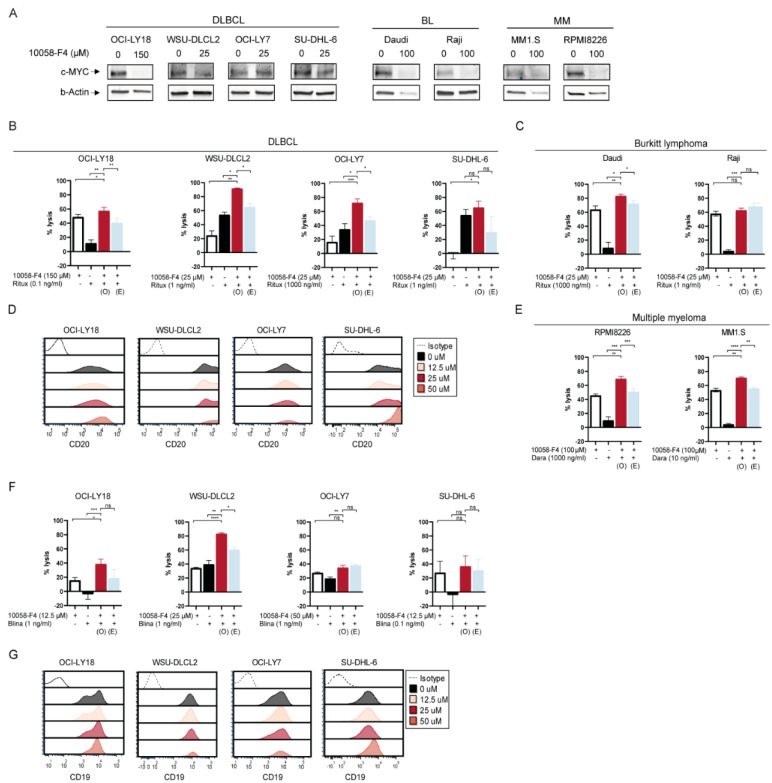
Combination of MYC inhibitor 10058-F4 with conventional and T cell-redirecting therapeutic antibodies increases tumor cell kill in various B-cell malignancies. (**A**) Immunoblot analysis of MYC protein and β-actin levels after treatment with indicated concentrations of 10058-F4 for 96 h (WSU-DLCL2, OCI-LY7 and SU-DHL-6), 48 h (OCI-LY18, Daudi, Raji and MM1.S) or 24 h RPMI8226. (**B**,**C**) Diffuse large B-cell lymphoma (DLBCL) cell lines (**B**) and Burkitt lymphoma (BL) cell lines (**C**) were pre-incubated with indicated concentrations of 10058-F4 (12.5–150 μM) for 96 h (WSU-DLCL2, OCI-LY7 and SU-DHL-6) or 48 h (Daudi and Raji) and thereafter incubated with predetermined concentrations of rituximab (ritux, 1–1000 ng/mL) in the presence of PBMCs obtained from healthy donors as effector cells (E:T ratio of 25:1) for 16 h. Tumor cell lysis was assessed using a bioluminescence-based cytotoxicity assay. Cell viability was calculated based on untreated cells. Observed [O] lysis upon combined treatment was compared to the expected [E] lysis, which was calculated according to the BLISS model (% expected lysis = ((% lysis 10058-F4/100) + (% lysis antibody/100)) − ((% lysis 10058-F4/100) × (% lysis antibody/100)) × 100). The null hypothesis of “additive effects” was rejected if the observed values were significantly different than the expected values. (**D**) Representative flow-cytometry histograms depict CD20 expression in DLBCL cell lines after 24 h of treatment with indicated concentrations (0–50 μM) of 10058-F4. Dotted line represents isotype staining control. (**E**) Multiple myeloma (MM) cell lines were pre-incubated with indicated concentrations of 10058-F4 (12.5–150 μM) for 48 h (MM1.S) or 24 h (RPMI8226) and thereafter incubated with predetermined concentrations of daratumumab (dara, 10–1000 ng/mL) in the presence of PBMCs obtained from healthy donors as effector cells (E:T ratio of 25:1) for 16 h. Details as in (**B**). (**F**) DLBCL cell lines were pre-incubated with indicated concentrations of 10058-F4 (12.5–150 μM) for 48 h (OCI-LY18) or 96 h (WSU-DLCL2, OCI-LY7 and SU-DHL-6) and thereafter incubated with predetermined concentrations of blinatumomab (blina, 0.1–1 ng/mL) in the presence of PBMCs obtained from healthy donors as effector cells (E:T ratio 10:1) for 24 h. Details as in (**B**). (**G**) Representative flow-cytometry histograms depict CD19 expression in DLBCL cell lines after 24 h of treatment with indicated concentrations (0–50 μM) of 10058-F4. Dotted line represents isotype staining control. Data are represented as mean ± SEM of 2–3 technical replicates. * *p* < 0.05; ** *p* < 0.01; *** *p* < 0.001; **** *p* < 0.0001; ns not significant.

**Figure 2 ijms-25-12094-f002:**
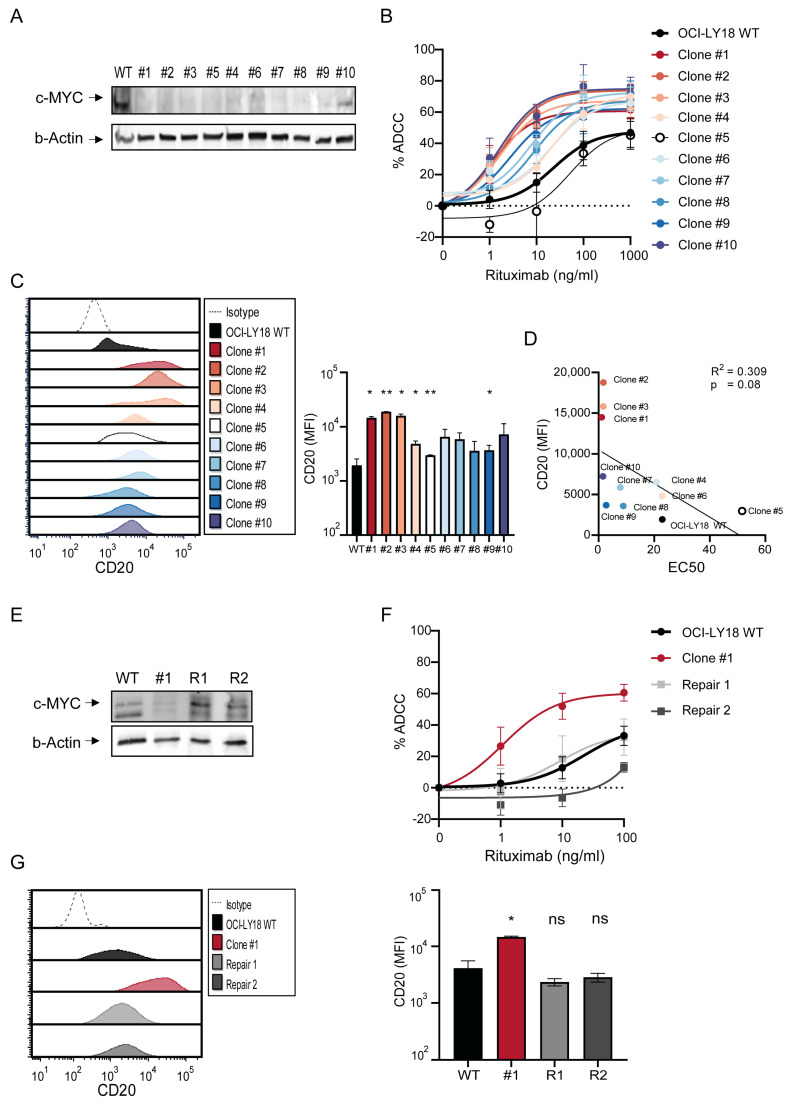
MYC downregulation increases rituximab-dependent cellular cytotoxicity and increases CD20 expression. (**A**) Immunoblot analysis of MYC protein and β-actin levels in OCI-LY18 wild type (WT) and MYC-targeted clones. Immunoblot is representative of three independent experiments. (**B**) OCI-LY18 WT (black) and MYC-targeted clones (colored) were incubated with indicated concentrations of rituximab (x-axis) in the presence of PBMCs obtained from healthy donors as effector cells (E:T ratio 40:1) for 16 h. Percentage of tumor cell lysis is depicted on the y-axis. Tumor cell lysis was assessed using a flow cytometry-based cytotoxicity assay. Cell lysis was calculated within each cell line compared to untreated cells. Data are presented as mean ± SEM of at least 3 technical replicates. (**C**) Representative flow-cytometry histogram depicting CD20 expression (left) and corresponding mean fluorescent values (MFI) (right) in OCI-LY18 WT (black) and MYC-targeted clones (colored). Expression data are representative histograms of at least 2 technical replicates. (**D**) Correlation plot of rituximab-induced EC50 values (x-axis) versus CD20 expression (as MFI values [y-axis]) in OCI-LY18 WT (black) and MYC-targeted clones (colored). (**E**) Immunoblot analysis of MYC protein and β-actin levels in OCI-LY18 WT and MYC-targeted clone #1 and MYC-repaired subclones (R1 and R2). Immunoblot is representative of three independent experiments. (**F**) Cytotoxicity of OCI-LY18 WT (black) and MYC-targeted clone #1 (red) and MYC-repaired subclones (grey) following rituximab treatment. Details as in (**B**). (**G**) Representative flow-cytometry histogram depicting CD20 expression (left) and corresponding mean fluorescent values (MFI) (right) in OCI-LY18 WT (black) and MYC-targeted clone #1 (red) and MYC-repaired subclones (grey). * *p* < 0.05; ** *p* < 0.01; ns not significant.

**Figure 3 ijms-25-12094-f003:**
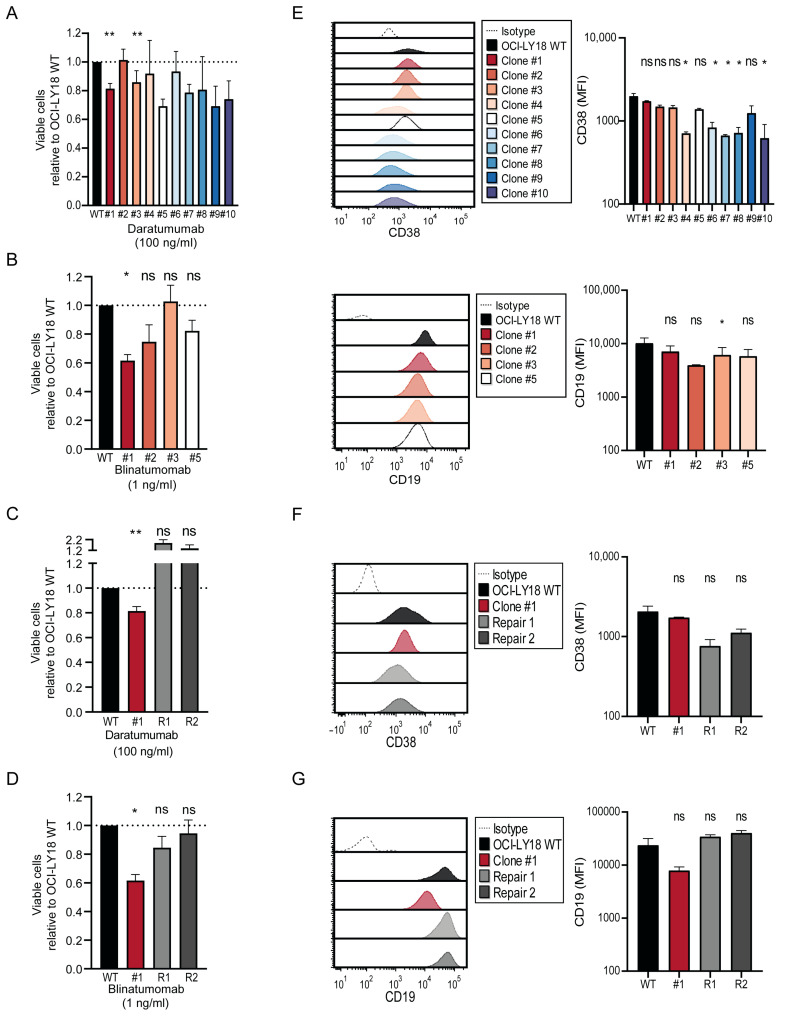
MYC downregulation increases lysis by conventional and T cell-redirecting therapeutic antibodies independent of target antigen upregulation. (**A**) OCI-LY18 wild type (WT, black) and MYC-targeted clones (colored) were incubated with 100 ng/mL daratumumab (x-axis) in the presence of PBMCs obtained from healthy donors as effector cells (E:T ratio 40:1) for 16 h. Percentage of tumor cell lysis is depicted on the y-axis. Tumor cell lysis was assessed using a flow cytometry-based cytotoxicity assay. Cell viability was calculated based on untreated cells (0 ng/mL), and the percentage of viable cells of the treated is normalized compared to OCI-LY18 WT. Data are presented as mean tumor cell lysis ± SEM of 3–4 independent experiments performed in duplicate. * indicates *p* < 0.05; ** *p* < 0.01, ns not significant. (**B**) OCI-LY18 WT (black) and a representative selection of MYC-targeted clones (colored) were incubated with 1 ng/mL blinatumomab in the presence of PBMCs obtained from healthy donors as effector cells (E:T ratio of 10:1) for 24 h. Details as in (**A**). (**C**) LY18 WT (black) and MYC-targeted clone #1 (red) and MYC-repaired subclones (grey) were incubated with 100 ng/mL daratumumab or blinatumomab in the presence of PBMCs obtained from healthy donors as effector cells (E:T ratio 40:1 (daratumumab) or 10:1 (blinatumomab)) for 16 h (daratumumab) or 24 h (blinatumomab). Details as in (**A**). (**D**) Representative flow-cytometry histograms depicting CD38 and CD19 expression in OCI-LY18 WT (black) and MYC-targeted clones (colored). (**E**–**G**) Representative flow-cytometry histograms depicting CD38 expression and CD19 expression (left) and corresponding mean fluorescent values (MFI) (right) in OCI-LY18 WT (black) and MYC-targeted clone #1 (red) and MYC-repaired subclones (grey). Expression data are representative histograms of at least 2 technical replicates.

**Figure 4 ijms-25-12094-f004:**
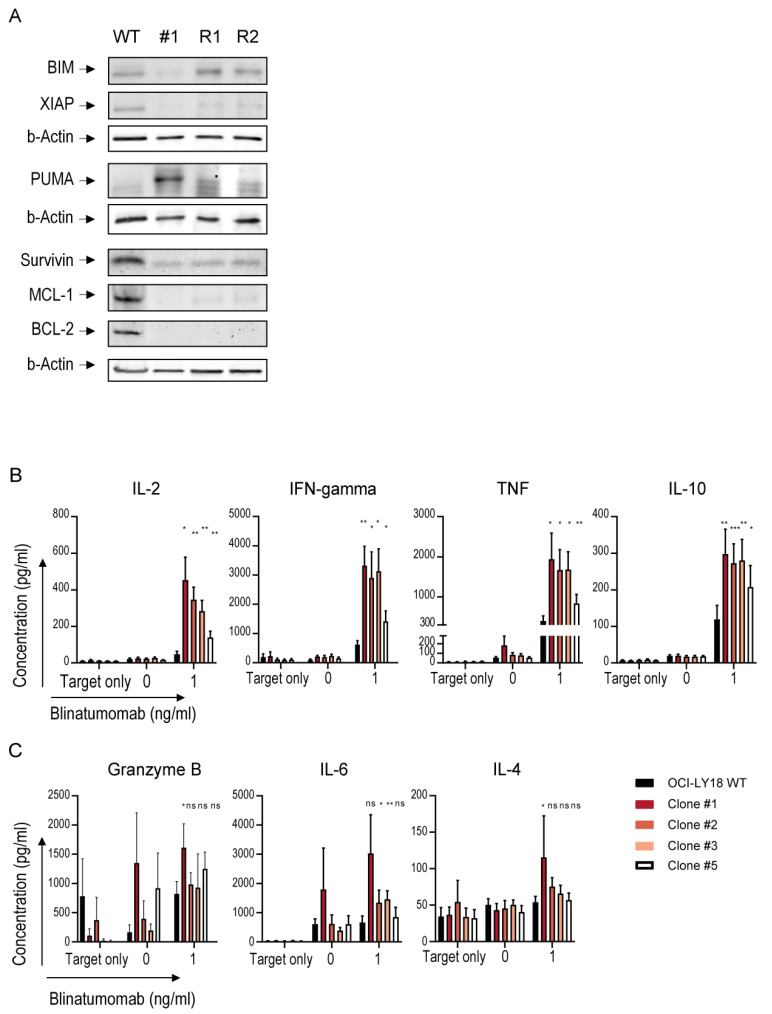
Mechanisms by which MYC mediates escape from conventional antibody and T cell-redirecting therapeutic antibodies. (**A**) Immunoblot analysis of BIM, XIAP, PUMA, Survivin, MCL-1 and BCL-2 and β-actin levels in OCI-LY18 wild type (WT) and MYC-targeted clone #1 and MYC-repaired subclones (R1 and R2). Immunoblot is representative of two independent experiments. (**B**,**C**) IL-2, IFN-γ, TNF, IL-10 (**B**) and Granzyme B, IL-6 and IL-4 (**C**) were measured in the cell culture supernatants of OCI-LY18 (black) and MYC-targeted clones (colored) treated with blinatumomab (0 and 1 ng/mL, x-axis) in the presence of PBMCs obtained from healthy donors as effector cells (E:T ratio 10:1) for 24 h. Target only represents the concentration of cytokines secreted in the absence of effector cells. The concentration of cytokines secreted in the supernatant (pg/mL) is depicted on the y-axis. Data represent mean cytokine secretion ± SEM of 3–4 independent experiments performed in duplicate. * *p* < 0.05; ** *p* < 0.01; *** *p* < 0.001; ns, not significant.

**Figure 5 ijms-25-12094-f005:**
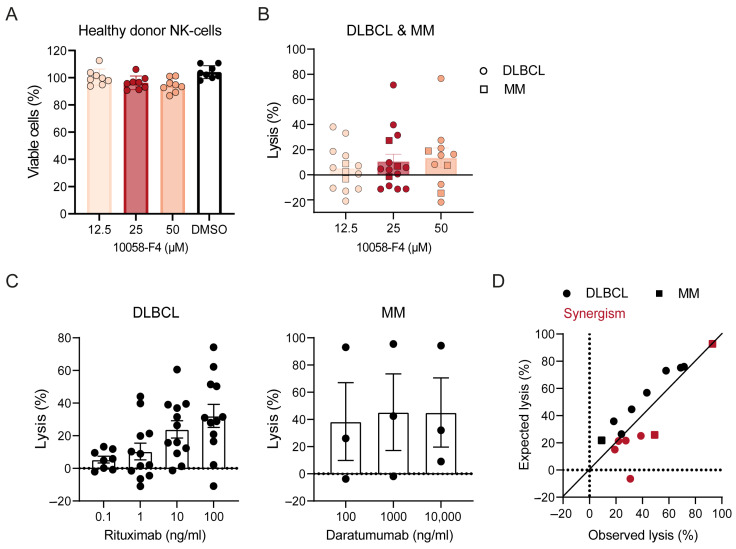
Combination of MYC inhibitor 10058-F4 with conventional therapeutic antibodies increases tumor cell kill in DLBCL and MM tumor samples. (**A**) Peripheral blood mononuclear cells from n = 8 healthy donors were incubated for 16 h with 0–50 μM 10058-F4 or DMSO control (x-axis). NK-cell viability was determined using flow cytometry-based cytotoxicity assay, and cell viability was calculated based on untreated cells (0 ng/mL). Error bars represent mean +/− SEM. (**B**) Diffuse large B-cell lymphoma (DLBCL) lymph node samples from n = 12 patients and multiple myeloma (MM) bone marrow samples from n = 3 patients were incubated with 0–50 μM 10058-F4 (x-axis) for 16 h. Details as in (**A**). (**C**) DLBLC lymph node samples from n = 12 and MM bone marrow samples from n = 3 patients were incubated with rituximab (in the case of DLBCL 0–1000 ng/mL) or daratumumab (in the case of MM, 0–10,000 ng/mL, x-axis) in the presence of PBMCs obtained from healthy donors as effector cells (E:T ratio 40:1) for 16 h. Tumor cell lysis was assessed using a flow cytometry-based cytotoxicity assay. Cell viability was calculated based on untreated cells. Bars and error bars represent mean tumor cell lysis ± SEM with mean lysis values of each sample (dots). (**D**) DLBCL samples from n = 12 patients (circles) and MM samples from n = 3 patients (squares) were incubated with 25 μM 10058-F4 and 100 ng/mL rituximab (DLBCL) or daratumumab (MM) in the presence of PBMCs obtained from healthy donors as effector cells (E:T ratio of 40:1) for 16 h. Tumor cell lysis was assessed using a flow cytometry-based cytotoxicity assay. Cell viability was calculated based on untreated cells. Observed [O] lysis upon combined treatment (x-axis) was compared to the expected [E] lysis (y-axis), which was calculated according to the BLISS model (% expected lysis = ((% lysis 10058-F4/100) + (% lysis antibody/100)) − ((% lysis 10058-F4/100) × (% lysis antibody/100)) × 100). The null hypothesis of “additive effects” was rejected if the observed values were significantly different than the expected values.

**Table 1 ijms-25-12094-t001:** Mutations in the sgRNA target sequences within the MYC gene of OCI-LY18 clones. CRISPR/Cas9-generated mutations in both MYC alleles of the individual OCI-LY18 clones. Depicted are nine individual clones following targeting with sgRNA-MYC-1, and clone #2 following targeting with sgRNA-MYC2. Both target sequences are underlined in the accompanying reference sequences. In the basepair column are the number of bases which are deleted(-) or added for each allele. For several clones, PCR insert cloning was conducted to determine the mutation for each allele, whereas for other clones, sequencing was conducted directly on the PCR product, and mutated sequences for each allele were deduced from the double peaks generated for each mutated base position.

Clone Nr	5′ *MYC-1* Target Site 3′	Basepair Nr
MYC1	CTCCCTTCGGGGAGACAACGACGGCGGTGGCGGGAGCTTCTCCACGGC	
Clone #1	CTCCCTTCGGGGAGACAACG.CGGCGGTGGCGGGAGCTTCTCCACGGC	−1
	CTCCCTTCGGGGAGACAACG......GTGGCGGGAGCTTCTCCACGGC	−6
Clone #3	CTCCCTTCGGGGAGACAAC..CGGCGGTGGCGGGAGCTTCTCCACGGC	−2
	CTCCCTTCGGGGAGACAAC......GGTGGCGGGAGCTTCTCCACGGC	−6
Clone #4	CTCC..................GGCGGTGGCGGGAGCTTCTCCACGGC	−18
	CTCCCTTCGG..................GGCGGGAGCTTCTCCACGGC	−18
Clone #5	CTCCCTTCGGGG....................GGAGCTTCTCCACGGC	−20
	CTCC.......................................ACGGC	−39
Clone #6	CTCCCTTCGGGGAGACAACG......GTGGCGGGAGCTTCTCCACGGC	−6
	CTCCCTTCGGGGGAGACAACGGGCGGCGGTGGCGGGAGCTTCTCCACGGC	2
Clone #7	CTCCCTTCGGGGAGACAACG......GTGGCGGGAGCTTCTCCACGGC	−6
	CTCCCTTCGG.................TGGCGGGAGCTTCTCCACGGC	−17
Clone #8	CTCCCTTCGGGGAGACAACG......GTGGCGGGAGCTTCTCCACGGC	−6
	CTCCCTTCGGGGAGACAAC..............GAGCTTCTCCACGGC	−14
Clone #9	CTCCCTTCGGGGAGACAACG......GTGGCGGGAGCTTCTCCACGGC	−6
	CTCCCTTCGG.................TGGCGGGAGCTTCTCCACGGC	−17
Clone #10	CTC...................GGCGGTGGCGGGAGCTTCTCCACGGC	−19
	CTC.......................................CACGGC	−39
MYC2	5′ *MYC-2* target site 3′	
	GGAACTATGACCTCGACTACGACTCGGTGCAGCCGTATTTCTACTGCG	
Clone #2	GGAACTATGACCTCGACTAC...TCGGTGCAGCCGTATTTCTACTGCG	−3
	GGAACTATGACCTCGACTAC..........AGCCGTATTTCTACTGCG	−10

**Table 2 ijms-25-12094-t002:** Repair of the MYC non-sense allele in OCI-LY18 clone 1. The original MYC-1 target site and the target site to induce homologous recombinant repair are shown in the reference sequence, above the CRISPR/Cas9-generated mutations in both *MYC* alleles of OCI-LY18 clone 1 (see also Table 1). Following CRISPR/Cas9-induced homologous recombination with the repair template, both *MYC* alleles in clone 1 were altered. All subsequent subclones derived from the “repaired” clone 1 had one *MYC* allele containing the identical sequence of the repair template and a second *MYC* allele without base deletions or additions. The full 129 base repair template is shown in the Section 4. The bases depicted in red differ from the wt *MYC* nucleotide sequence, since this template design prevents perpetual CRISPR/Cas9 nuclease activity but still allows the generation of the wt MYC protein in the cells upon successful recombination. Respectively, both allele sequences translate to PPLSPSRRSGLCSPSYVAVTPFSLRGDNDGGGGSFSTADQL, which is MYC wt, and to PPLSPSRRSGLCSPSYGCGHTLLPSGKQEGGGGSFSTADQLE. In the basepair column are the number of bases shown which are deleted(-) for each *MYC* allele. For several subclones, PCR insert cloning was conducted to determine the mutation for each allele, whereas for other subclones, sequencing was conducted directly on the PCR product, and mutated sequences for each allele were deduced from the double peaks generated for each mutated base position.

Clones	5′ *MYC*-1 Target Site 3′	Basepair Nr
	CCGGGCTCTGCTCGCCCTCCTACGTTGCGGTCACACCCTTCTCCCTTCGGGGAGACAACGACGGCGGTGGCGGGA	
Clone 1	CCGGGCTCTGCTCGCCCTCCTACGTTGCGGTCACACCCTTCTCCCTTCGGGGAGACAACG.CGGCGGTGGCGGGA	−1
	CCGGGCTCTGCTCGCCCTCCTACGTTGCGGTCACACCCTTCTCCCTTCGGGGAGACAACG......GTGGCGGGA	−6
All subclones Following repair	CCGGGCTCTGCTCTCCCTCTTACGTCGCTGTCACACCCTTCTCCCTTCGGTGAGATAATGATGGCGGTGGCGGGA CCGGGCTCTGCTCGCCCTCCTACGGTTGCGGTCACACCCTTCTCCCTTCGGGAAAACAAGAAGGCGGTGGCGGGA	

## Data Availability

All data are available in the main text or the Appendix A. For source data, please contact a.dejonge1@amsterdamumc.nl.

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
