# Peer review of "Delineating MYC-Mediated Escape Mechanisms from Conventional and T Cell-Redirecting Therapeutic Antibodies"

_ijms, 2024, doi:10.3390/ijms252212094_

Round 1

Reviewer 1 Report

Comments and Suggestions for Authors

In their article "Delineating MYC-mediated escape mechanisms from conventional and T-cell redirecting therapeutic antibodies" de Jonge et al. investigate the effect of MYC overexpression on antibody mediated T cell and NK cell cytotoxicity. With extensive in vitro experiments the authors demonstrate two different pathways how MYC overexpression can affect susceptibility to antibody mediated therapies. The article is well written, coherent and within the scope of the journal. There are no major points and I recommend acceptance after the following minor points have been addressed:

- harmonize fonts in abstract and Title of Table 2

- please clarify with the journal - to my opinion the methods from the supplement belong into the manuscript except of the primer and plasmid specificities.

- Suppl Fig 1 A and B: legends are swapped

- Fig. 1A shows Daudi/Raji incubated with 100uM 10054-F4, Fig 1B 25uM - please explain why the concentration was changed

- lines 295 and 300: " The bases depicted in red differ from the wt MYC nucleotide" and " The amino acids in red differ from the MYC wt" are redundant. Skip one.

- line 305 section 3.5. is colored in red. Change to black

- line 311 Figure S2F follows on S3A-B, move it to S3C to keep the order

- Figure 4 B and C: it’s unclear what target, 0, 1 stands for. According to the legend its 0-10ng/ml Blina.

- Fig 5B: some circles are darker

- 5E is in the legend but not shown in the graph

- discuss the mechanism how MYC can (potentially) regulate CD20 expression

Author Response

General response from the authors: We sincerely thank the reviewer for carefully reading the manuscript. We apologize for the inconsistencies and unintended mistakes in legend orders. We have adjusted them according to the reviewer’s suggestion.

Comment 1: harmonize fonts in abstract and Title of Table 2

Response 1: We have now harmonized the fonts in the abstract (page 1) and title of Table 2 (page 8, line 255).

Comment 2: please clarify with the journal - to my opinion the methods from the supplement belong into the manuscript except of the primer and plasmid specificities.

Response 2:  We agree with the reviewer that this is important information. Because of the manuscript's length, we included it in the supplementary materials. If the editor prefers this information to be in the main text, we will adjust it accordingly.

Comment 3: Suppl Fig 1 A and B: legends are swapped

Response 3: We have placed the legends in the right order (Supplementary file, page 13).

Comment 4: Fig. 1A shows Daudi/Raji incubated with 100uM 10054-F4, Fig 1B 25uM - please explain why the concentration was changed

Response 4:  We reduced the concentration because 100 uM was found to be too toxic in cytotoxicity experiments. We did not want to achieve 100% lysis with 10058-F4 alone, as this would hinder our ability to observe the effects of the combination with rituximab more effectively.

Comment 5: lines 295 and 300: " The bases depicted in red differ from the wt MYC nucleotide" and " The amino acids in red differ from the MYC wt" are redundant. Skip one.

Response 5: We agree that this information is redundant and we have deleted the sentence “the amino acids in red differ from the MYC wt” (page 9, line 300).

Comment 6: line 305 section 3.5. is colored in red. Change to black

Response 6: We believe that this mistake was generated while formatting the manuscript to the IMSJ format by the editor. We have changed the color of “3.5”  to black (page 9, line 305).

Comment 7: line 311 Figure S2F follows on S3A-B, move it to S3C to keep the order

Response 7: We have adjusted the order of the figures according to the reviewer’s suggestion. What was previously Figure S3C is now Figure S3D (page 9-10, lines 311-321).

Comment 8:  Figure 4 B and C: it’s unclear what target, 0, 1 stands for. According to the legend its 0-10ng/ml Blina.

Response 8: We agree that this could be better clarified. We have adjusted the legend as follows: “… treated with blinatumomab (0 and1 ng/ml, x-axis) in the presence of PBMCs obtained from healthy donors as effector cells (E:T ratio 10:1) for 24h. Target only represents concentration of cytokines secreted in absence of effector cells. Concentration of cytokines secreted in the supernatant (pg/ml) is depicted on the y-axis.”

We have changed the x-axis of the figure to “Target only”, “0” and “1” (page 13, lines 379-380) and replaced the figure in the text document. We hope the figure is now more clear.

Comment 9:  Fig 5B: some circles are darker

Response 9: We have adjusted the line with of the circles and replaced the figure in the text document (page 15, line 412). We apologize for the inconsistency.

Comment 10:  5E is in the legend but not shown in the graph

Response 10:  We have deleted 5E in the legend (page 15, lines434-436). We apologize for the mistake.

Comment 11: discuss the mechanism how MYC can (potentially) regulate CD20 expression

Response 11: We agree with the reviewer that this could be further discussed. We have added the following text on page 16,  lines 474-476:

“It was was previously suggested that MYC directly represses the promotor of the MS4A1 gene, encoding CD20, and via induction of the MS4A1/CD20-targeting miR-222.”

Reviewer 2 Report

Comments and Suggestions for Authors

In this manuscript, the authors addressed an important topic on the effects of MYC overexpression with regards to immunotherapy response. The authors conducted several experiments to show how MYC inhibition (using 10058-F4) could potentially enhance immunotherapy response. While previous studies have hinted at the effects of MYC inhibition in improving immunotherapy outcomes, this manuscript provides a deeper exploration.

In this manuscript, CRISPR technology was employed to target and modify the MYC gene in B-cell malignancies, enabling the authors create MYC downregulated clones. This allowed for a detailed study of MYC's role in immune evasion and resistance to immunotherapy. By generating and analyzing these clones, the authors observed the direct effects of MYC downregulation, such as increased susceptibility to antibody-mediated cytotoxicity and changes in apoptotic pathway proteins. Repairing the MYC gene in the clones validated the observed effects, concluding they are primarily due to MYC downregulation. This study I believe provides insight into the potential therapeutic strategies that could be applied to possibly enhance the efficacy of immunotherapies in MYC-driven cancers.

In my opinion the manuscript is well written and contributes important knowledge to the field. The methods and results are presented in detail, and the authors effectively communicate their findings.

Author Response

We thank the reviewer for his/her valuable comments and putting trust in our work.
